# Multilevel analysis of dropout from maternal continuum of care and its associated factors: Evidence from 2022 Tanzania Demographic and Health Survey

Angwach Abrham Asnake[1]*, Amanuel Alemu Abajobir[2,3], Beminat Lemma Seifu[4], Yordanos Sisay Asgedom[1], Molalgn Melese[5], Meklit Melaku Bezie[6], Yohannes Mekuria Negussie[7]

1 Department of Epidemiology and Biostatistics, School of Public Health, College of Medicine and Health Sciences, Wolaita Sodo University, Wolaita Sodo, Ethiopia, 2 African Population and Health Research Center, Nairobi, Kenya, 3 School of Public Health, The University of Queensland, Brisbane, Australia, 4 Department of Public Health, College of Medicine and Health Sciences, Samara University, Samara, Ethiopia, 5 Department of Reproductive Health and Nutrition, School of Public Health, College of Medicine and Health Sciences, Wolaita Sodo University, Wolaita Sodo, Ethiopia, 6 Department of Public Health Officer, Institute of Public Health, College of Medicine and Health Sciences, University of Gondar, Gondar, Ethiopia, 7 Department of Medicine, Adama General Hospital and Medical College, Adama, Ethiopia

* angwachabrham@gmail.com

## Abstract

### Background

The maternal continuum of care (CoC) is a cost-effective approach to mitigate preventable maternal and neonatal deaths. Women in developing countries, including Tanzania, face an increased vulnerability to significant dropout rates from maternal CoC, and addressing dropout from the continuum remains a persistent public health challenge.

### Method

This study used the 2022 Tanzania Demographic and Health Survey (TDHS). A total weighted sample of 5,172 women who gave birth in the past 5 years and had first antenatal care (ANC) were included in this study. Multilevel binary logistic regression analyses were used to examine factors associated with dropout from the 3 components of maternal CoC (i.e., ANC, institutional delivery, and postnatal care (PNC)).

### Results

The vast majority, 83.86% (95% confidence interval (CI): 82.83%, 84.83%), of women reported dropout from the maternal CoC. The odds of dropout from the CoC was 36% (AOR = 0.64, (95% CI: 0.41, 0.98)) lower among married women compared to their divorced counterparts. Women who belonged to the richer wealth index reported a 39% (AOR = 0.61, (95% CI: 0.39, 0.95)) reduction in the odds of dropout, while those belonged to the richest wealth index demonstrated a 49% (AOR = 0.51, (95% CI: 0.31, 0.82)) reduction. The odds of dropout from CoC was 37% (AOR = 0.63, (95% CI: 0.45,0.87)) lower among women who

**Data Availability Statement:** Data is available from https://www.dhsprogram.com.

**Funding:** The author(s) received no specific funding for this work.

**Competing interests:** The authors have declared that no competing interests exist.

reported the use of internet in the past 12 months compared to those who had no prior exposure to the internet. Geographical location emerged as a significant factor, with women residing in the Northern region and Southern Highland Zone, respectively, experiencing a 44% (AOR = 0.56, 95% CI: 0.35–0.89) and 58% (AOR = 0.42, 95% CI: 0.26–0.68) lower odds of dropout compared to their counterparts in the central zone.

## Conclusion

The dropout rate from the maternity CoC in Tanzania was high. The findings contribute to our understanding of the complex dynamics surrounding maternity care continuity and underscore the need for targeted interventions, considering factors such as marital status, socioeconomic status, internet usage, and geographical location.

## Introduction

The maternal continuum of care (CoC) is the provision of healthcare spanning from pregnancy to childbirth and into the postpartum period. This includes antenatal care (ANC) during pregnancy, the attendance of a skilled birth attendant during delivery, and the receipt of postnatal care (PNC) following delivery [1–3]. The maternity CoC emerges as a fundamental program strategy to reduce maternal and newborn mortality rates and enhance the health outcomes and wellbeing of mothers and neonates [3–6]. The maternity CoC is a simple and cost-effective strategy that reduces preventable maternal and neonatal deaths, promoting optimal health for women and neonates [7, 8], and appropriate CoC is predicted to curb half a million maternal deaths, 4 million neonatal deaths, and 6 million deaths of children [9]. It highlights the significance of links across maternity care service packages provided during pregnancy, childbirth, and the postpartum period, alongside the home, primary, secondary, and tertiary levels of care in healthcare delivery [10–12]. A strong CoC integrates maternity, neonatal, and child health (MNCH) services during pregnancy, delivery, and postpartum periods [10]. However, poor MNCH outcomes are associated with a lack of proper care at any phase of a CoC [11, 12].

Globally, there was a 34% decrease in the maternal mortality ratio between 2000 and 2020. As of 2020, the maternal mortality ratio stood at 223 deaths per 100,000 live births, slightly lower than the 2015 figure of 227 deaths [13]. Despite this progress, the threat of pregnancy-related preventable morbidity and mortality persisted at elevated levels. In 2020 alone, an estimated 287,000 women died globally from preventable causes linked to pregnancy and childbirth, equivalent to 800 maternal deaths each day [13, 14]. Maternal and child mortality disproportionately impacts women and children in low and lower-middle incomes (LMICs). Sub-Saharan Africa (SSA), contributing to around 70% of global maternal deaths in 2020, bears a significant burden. Despite significant improvements, Tanzania continues to endure one of the highest rates of maternal and child mortality in the region [13–15].

Dropout from the CoC refers to the failure to adhere to recommended maternity care protocols, such as attending a minimum of 4 ANC visits, delivering at a health institution, and receiving at least one postnatal check-up within 6 weeks postpartum [16]. Specifically, discontinuation of care before completing the recommended number of ANC visits (at least 4) constitutes dropout from ANC [17]. Dropout from institutional delivery occurs when women give birth outside of health facilities [18]. Similarly, failure to undergo at least 1 postnatal check-up within 6 weeks after delivery categorizes a mother as a PNC dropout [16].

Based on research conducted across 28 African countries, 44% of women experienced discontinuation of maternal CoC [19]. In SSA, three-quarters (75%) of women were found to be CoC dropouts [20]. A study in Colombia showed 95% of women were dropouts. Similarly, in Ethiopia, the proportion of women experiencing CoC dropout was reported at 87.1% [21]. Only about 10.7% Ugandan women received complete CoC [16]. Additionally, in Tanzania, merely 10% of women adhered to the recommended visit schedule throughout the continuum [22]. Overall, women in developing countries, including Tanzania, face a heightened susceptibility to substantial dropout rates from the maternal CoC, both transitioning from ANC to institutional delivery and progressing from institutional delivery to PNC services [1, 7, 23]. This suggests that most women did not receive the health benefits that the CoC provides for both the mother and the child. Maternal age, place of residence, marital status, religion, educational status, parity, wealth index, media exposure, and distance to health facilities were identified as different factors that contribute to the dropout from the maternal CoC [1, 7, 18, 19, 22, 23].

Although there are some studies in Tanzania that assess factors associated with dropout from CoC, they did not address dropouts from the 3 components of CoC separately, nor combined CoC index. It is important to analyze each component of CoC separately to identify areas of high dropouts. While combined CoC promotes comprehensive and systematic understanding of MNCH, discrete CoC components reflect the uptake of, and dropout from, specific services within the continuum and to design tailored interventions to enhance uptake of all essential services. Maternal CoC also integrates both temporal and spatial aspects, reflecting when and where care is provided. This study was conducted to assess the magnitude of dropout from the maternal CoC and its associated factors after ANC booking among reproductive age women in Tanzania, using the recent data from 2022 Tanzanian Demographic and Health Survey (TDHS).

## Method

### Data source, study setting, and design

This study used the 2022 Tanzania Demographic and Health Survey (TDHS) which is available and accessed from http://www.dhsprogram.com. The TDHS uses a cross-sectional design. The sample design for the 2022 TDHS had 2 stages and was intended to provide estimates for the entire country, for urban and rural areas on the mainland, and for Zanzibar. The first stage involved selecting sample points (clusters), consisting of enumeration areas (EAs) delineated for the 2012 Tanzania Population and Housing Census (PHC). A total of 629 clusters were selected. Among the 629 EAs, 211 EAs were from urban areas and 418 EAs were from rural areas. In the 2nd stage, 26 households were to be systematically selected from each cluster, for a total anticipated sample size of 16,354 households for the 2022 TDHS [24].

### Study participants and sample size

A total of 15,254 women were interviewed for maternal health and 7,681 women who gave birth the past 5 years before preceding the survey were interviewed for ANC visits and place of birth. A total weighted sample of 5,172 women who gave birth in the past 5 years and had first antenatal bookings were included in this study.

### Dependent variable and its operational definition

The outcome variable for this study was a dropout from the maternal CoC. Dropout from ANC visits was considered if women had no at least 4 ANC visits after booking for ANC

service [18]. It was generated by recoding the number of less than 4 ANC visits "1," representing dropout from ANC, and those with 4 and more visits as "0," indicating complete ANC. Dropout from institutional delivery was considered if women gave birth out of health facilities after ANC booking [18]. It was recoded those delivered in public and private facilities as "0," which indicate institutional delivery and those delivered else recoded as "1," which indicates dropout from institutional delivery. If the participants had no at least 1 PNC visit, this variable was recoded as "1," indicating dropout from PNC, otherwise it was recoded as "0". Maternal CoC was considered when a woman had at least 4 ANC visits, had delivered in a health institution, and had at least 1 PNC check-up within 6 weeks after childbirth [16]. This helps identify dropouts from specific CoC components for tailored interventions. Conversely, if they missed 1 or more of the three CoC components, they were recoded as "1," indicating dropout from CoC. While the maternal CoC encompasses various components, most of these components are provided within healthcare facilities during the pivotal stages of ANC, childbirth, and PNC [9]. The incorporation of all these 3 components in the formulation of the CoC variable ensures that essential preventive care services are comprehensively administered during these critical points of contact.

### Independent variables

Both individual- and community-level variables were taken into consideration. Individual-level variables were age of the respondents, age at first birth, parity, marital status, educational status, sex of household head, wealth index, media exposure, internet use, and health insurance. Community-level variables were place of residence, geographic region, distance to health facilities, community illiteracy, community wealth index, and community media exposure.

Media exposure was operationalized as the summation of exposures to radio and television, with individuals using at least 1 of these mediums considered to have media exposure. Community illiteracy and community wealth index were derived by aggregating individual-level variables at the cluster level. Community illiteracy was calculated by dividing the number of individuals with no education by the total population, with those above the median proportion considered to have high community illiteracy since the proportion does not have a normal distribution. Similarly, community wealth status was generated with "poor" as the reference category. Finally, variables such as place of residence, geographic region, distance to health facilities, community illiteracy, and community wealth index were deemed community-level variables and incorporated into the analysis.

### Statistical analysis and model building

The data were analyzed by using STATA version 17. Descriptive statistics were presented using tables and figures. A sampling weight was applied in each analysis taking the nonproportional distribution of the sample between urban and rural locations into consideration as well as any differences in response rates. Proper representativeness of the study was thus ensured both nationally and regionally. To account for the clustering effects of DHS data, a multilevel binary logistic regression model was applied to determine the effect of each independent variable on dropout of from maternal CoC. Both bivariable and multivariable multilevel analysis were conducted, and variables with a p-value less than 0.20 in this analysis, and those relevant from the literature were considered as candidates for multivariable multilevel binary logistic regression analysis.

To evaluate heterogeneity among clusters, we calculated the Likelihood Ratio (LR) test, Intra-class Correlation Coefficient (ICC), and Median Odds Ratio (MOR). The ICC is a measure of the relatedness of clustered data. It accounts for the relatedness of clustered data by

comparing the variance within clusters with the variance between clusters [25].

$$\text{ICC} = \frac{s_b^2}{s_b^2 + s_w^2}$$

Where $s_b^2$ = variance between clusters, $s_w^2$ = variance within the cluster.

The MOR measures the variation in the dropout from the CoC when we randomly select 2 reproductive age women during data collection from 2 clusters that had high- and low-risk of dropout rate from the CoC [26].

$$\text{MOR} = \exp\sqrt{(2*\partial^2*0.6745)} \sim \text{MOR} = \exp(0.95*\partial).$$

$\partial^2$ indicates cluster variance.

In bi-variable multilevel binary logistic regression analysis variables with a p-value < 0.2 were considered for the multivariable analysis. Four models were constructed for the multivariable multilevel binary logistic regression. The first model was a null model without any explanatory variables, the 2nd model was fitted with individual-level variables, the 3rd with community-level variables, and the 4th with both individual and community-level variables at the same time. The fitted model was compared using deviance and the lowest deviance was considered as the best-fit model. Finally, the Adjusted Odds Ratio (AOR) of the good fitted model with its 95% confidence interval (CI) was reported, and variables were considered statistically significant if their p-value < 0.05 in the multivariable analysis.

## Ethical consideration

This study did not need participant permission or ethical clearance because it utilized secondary (TDHS) data. The International Consulting and Fulfillment's (ICF's Inc.) Ethical Review Board reviewed and approved the survey protocol, including questionnaires, of all DHSs. After submitting the proposal of this study, we have obtained written permission to download and use the dataset for 2022 TDHS from ICF and MEASUREDHS at http://www.dhsprogram.com. The obtained dataset was anonymous. The 2022 TDHS report contained all the necessary materials regarding ethical standards of the survey.

## Results

### Characteristics of the participant

Out of a weighted sample of 5,172 participants, 3,686 (71.26%) were from rural areas with a median age of 28 (IQR: 23–34) years. Majority, 4406 (85.19%), of the participants were married. Around one-fifth, 19.21% (of women were uneducated. Approximately half, 49.26% (2462), of the study participants had no media exposure and the vast majority, 87.55% (2462), of women had never used internet. More than half, 58.05% (3002), women were from a highly illiterate community (Table 1).

### The magnitude of dropout from maternal continuum of care

The vast majority, 83.86% (95% CI: 82.83%, 84.83%), of women dropout from maternal CoC. More than one-fourth, 27.13% (95% CI: 25.93, 28.36), of women dropout from ANC, 16.55% (95% CI: 15.56%, 17.59%), while dropout from institutional delivery and PNC was noted at 16.55% (95% CI: 15.56%, 17.59%) and 82.81% (95% CI: 81.75%, 83.81%), respectively (Fig 1).

More than one-fourth, 1,133 (26.77%), of the participants who experienced dropout from ANC were married, with approximately 730 (17.25%) encountering dropout during institutional delivery and 3,512 (82.97%) of these women exhibited dropout from PNC.

**Table 1. Background characteristics of respondents reproductive age women in Tanzania (n = 5,172).**

| Variable name | Categories | Weighted frequency | Percentage (%) |
|---|---|---|---|
| Age (in years) | 15–19 | 826 | 8.03 |
| | 20–24 | 1344 | 25.98 |
| | 25–29 | 1316 | 25.45 |
| | 30–34 | 963 | 18.62 |
| | 35–39 | 706 | 13.65 |
| | 40–44 | 348 | 6.73 |
| | 45–49 | 70 | 1.34 |
| Parity | 1–2 | 2296 | 44.39 |
| | 3–4 | 1602 | 30.97 |
| | ≥5 | 1275 | 24.64 |
| Age at first birth | <18 | 2279 | 44.06 |
| | ≥18 | 2894 | 55.94 |
| Marital status | Single | 441 | 8.52 |
| | Married | 4406 | 85.19 |
| | Widowed | 48 | 0.93 |
| | Divorced | 278 | 5.37 |
| Wealth index | Poorest | 994 | 19.21 |
| | Poorer | 1083 | 20.93 |
| | Middle | 1011 | 19.55 |
| | Richer | 1004 | 19.41 |
| | Richest | 1081 | 20.90 |
| Educational status | Uneducated | 1002 | 19.38 |
| | Primary | 2888 | 55.83 |
| | Secondary and higher | 1282 | 24.79 |
| Media exposure | No | 2462 | 49.26 |
| | Yes | 2536 | 50.74 |
| Internet use | Never | 4528 | 87.55 |
| | In last 12 months | 555 | 10.74 |
| | Before 12 months | 89 | 1.71 |
| Health insurance | No | 4918 | 95.08 |
| | Yes | 255 | 4.92 |
| Place of residence | Urban | 1487 | 28.74 |
| | Rural | 3686 | 71.26 |
| Distance to health facilities | Not big problem | 1712 | 33.11 |
| | Big problem | 3460 | 66.89 |
| Community illiteracy | Low illiteracy | 2170 | 70.95 |
| | High illiteracy | 3002 | 58.05 |
| Community wealth index | Low poverty | 2227 | 43.06 |
| | High poverty | 2945 | 56.94 |
| Community media exposure | Low media exposure | 2799 | 54.12 |
| | High media exposure | 2373 | 45.88 |

Approximately 439 (40.62%) of economically disadvantaged women experienced dropout from ANC, with 370 (34.26%) and 928 (85.90%) encountering dropout during institutional delivery and PNC after ANC booking, respectively.

Among women without media exposure, around one-third, 793(32.21%), experienced dropout from ANC, while more than one-fifth, 550(22.62%), faced dropout during

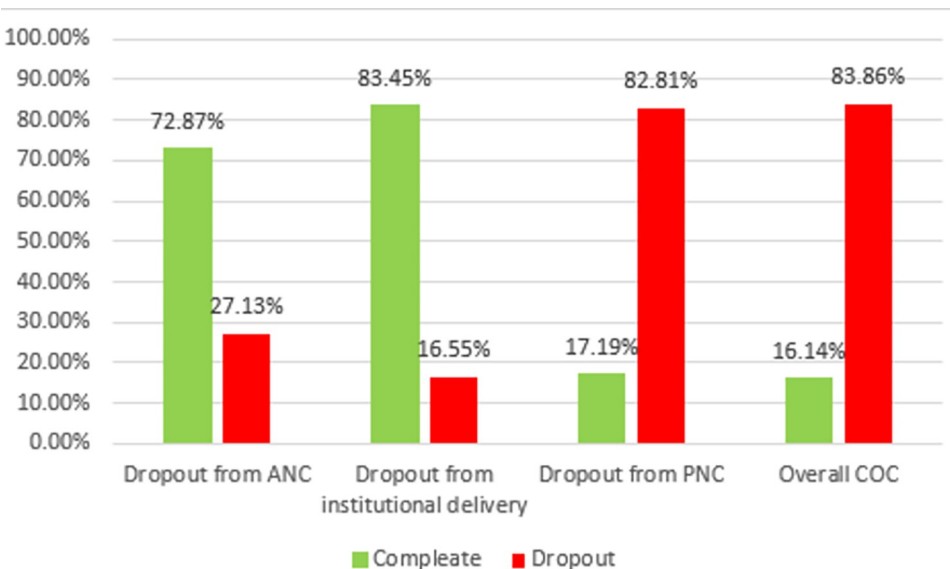

**Fig 1. Prevalence of dropout from the continuum of maternity care services among Tanzanian reproductive age women, 2022.**

institutional delivery and 2,103(85.44%) experienced dropout from PNC following ANC booking (S1 Table).

## Random effect and model comparison

The ICC in the null model for dropout from maternal CoC was 0.17, which means that 17.02% of the variability in dropout from CoC was due to differences between clusters or unobserved factors at community level. According to the ICC calculated based on estimated intercept component variance, approximately 17.08% of the variation in the odds of dropout from CoC in women could be attributed to community-level factors. After adjusting for individual- and community-level factors, the variation in dropout in the continuum of care remained statistically significant. The full model (model 4) showed that about 12.88 across communities were observed. The MOR in all models was greater than 1, which indicates that there is a variation in the dropout of 3 components of CoC between community levels (clusters). Moreover, the MOR for dropout from CoC, antenatal care, institutional delivery, and PNC in the null model was 1.95, 2.19, 5.27, and 2.14 respectively, indicating that there was variability between clusters. The difference in dropout rates of ANC, institutional delivery, PNC, and CoC between communities was found to be statistically significant with a p-value of less than 0.001 (Table 2 and S2 Table). If we randomly selected individuals from 2 different clusters, those in the cluster

**Table 2. Random effect results and model comparisons.**

| Measure of variation for dropout from CoC | | | | |
|---|---|---|---|---|
| | Null model | Model 1 | Model 2 | Model 3 |
| Community level variance (95% CI) | 0.68 (0.52, 0.88) | 0.60 (0.42, 0.87) | 0.52 (0.35, 0.76) | 0.49 (0.32, 0.73) |
| p-value | < 0.0001 | < 0.0001 | < 0.0001 | < 0.0001 |
| Deviance | 4261.58 | 4055.34 | 4201.90 | 4,007.50 |
| ICC % | 17.20% | 15.46% | 13.53% | 12.88% |
| MOR | 2.18 | 2.10 | 1.99 | 1.95 |

with a higher dropout from CoC had 2.18 times the odds of dropout from CoC compared to those in the cluster with a lower risk of dropout from CoC (Table 2). The smaller deviance value is in the 4th model which includes both the individual- and community-level factors of ANC, institutional delivery, PNC, and overall CoC (Table 2 and S2 Table).

### Factors associated with dropout from maternal continuum of care

This study found that both individual- and community-level factors were significantly associated with dropout from the 3 components of maternal CoC separately and the overall CoC in Tanzanian reproductive age women. The odds of dropout from CoC was 36% (AOR = 0.64, (95% CI: 0.41, 0.98)) lower among married women compared to divorced women. Women who belonged to a richer and richest wealth index had a 59% (AOR = 0.61, (95% CI: 0.39–0.95)) and 49% (AOR = 0.51, (95% CI: 0.31,0.82)) reduced odds of dropout from CoC. The odds of dropout from CoC was 37% (AOR = 0.63, (95% CI: 0.45,0.87)) lower in women who used internet in the past 12 months compared to those who never exposed to internet. In comparison with women were living in central Zone, women who resided in the Northern region and Southern Highland Zone reported 44% (AOR = 0.56, (95% CI: 0.35–0.89)) and 58% (AOR = 0.42, (95% CI: 0.26–0.68)) lower odds of dropout from CoC, respectively (Table 3).

The analyses into factors contributing to dropout rates from ANC, institutional delivery and PNC separately, among Tanzanian reproductive age women showed significant associations. In terms of ANC dropout, age played a pivotal role, with increasing odds noted in age groups 25–29, 30–34, 35–39, 40–44, and 45–49. Unmarried status increased the odds of ANC dropout by 1.37 times. Economic status also influenced ANC dropout, as the wealthiest women exhibited a 68% reduction in odds of dropout. Grand multiparity was associated with 1.78 times increase in ANC dropout, and distance to a health facility raised the odds by 1.32 times. Internet use in the past 12 months demonstrated a significant association, with a 46% reduction in odds of ANC dropout. Health insurance was associated with a 47% reduction in odds of ANC dropout. For institutional delivery, being in the richest wealth index reduced the odds by 74%, while age group 40–44, higher education, rural residence, and internet use decreased the odds of dropout. Grand multiparity, however, increased the odds by 3.35 times. Health insurance was associated with a 35% reduction in odds of PNC dropout. Media exposure and internet use were associated with 24% and 32% lower odds of dropout from PNC, respectively (S3 Table).

## Discussion

This study aimed to measure the level of maternal CoC and to identify factors associated with dropout from the CoC using the recent nationally representative data from the TDHS. A high rate (83.86%) of dropout from maternal CoC. The largest dropout was observed in PNC service. The study, by employing a multilevel analysis, revealed both individual- and community-level factors attributed to this dropout. Employing multilevel analysis enabled us to account for the variation of maternal CoC across clusters.

The overall dropout rate of maternal CoC is lower than the rates reported in previous studies conducted in Ethiopia [18], Tanzania [22], Uganda [16], Gambia, and Ghana [27], but higher than reports from a study conducted in Kenya [28]. There could be various reasons for this discrepancy, including variations in sampling and analysis approaches. For instance, studies in in Tanzania, Uganda, Gambia, and Ghana included women who reported any ANC visit (not first ANC), but the current study excluded those women without first ANC, which may lead to overestimation of the rates. Other possible reasons might be due to sociocultural differences in the study population.

**Table 3. Multilevel analysis of factors associated with dropout from maternal CoC among reproductive age women in Tanzania.**

| Variables | Categories | Dropout from CoC AOR (95% CI) | | |
|---|---|---|---|---|
| | | Model 1 | Model 2 | Model 3 |
| | | (Individual-level variables) | (Community-level variables) | (Both community and individual level variables) |
| Age (in years) | 15–19 | 1.00 | | 1.00 |
| | 20–24 | 0.72 (0.50, 1.03) | | 0.71 (0.94, 1.02) |
| | 25–29 | 0.79 (0.53, 1.17) | | 0.81 (0.54, 1.20) |
| | 30–34 | 0.83 (0.53, 1.29) | | 0.85 (0.54,1.20) |
| | 35–39 | 0.94 (0.58, 1.52) | | 0.95 (0.58,1.53) |
| | 40–44 | 0.91 (0.53, 1.56) | | 0.92 (0.54, 1.60) |
| | 45–49 | 0.94 (0.43, 2.03) | | 0.94 (0.43,2.04) |
| Parity | 1–2 | 1.00 | | 1.00 |
| | 3–4 | 1.01 (0.79, 1.28) | | 0.95 (0.75, 1.26) |
| | ≥5 | 0.82 (0.59, 1.14) | | 0.78 (0.56, 1.10) |
| Media exposure | No | 1.00 | | 1.00 |
| | Single | 0.57 (0.34, 0.96)* | | 0.65 (0.39,1.12) |
| | Married | 0.61 (0.39, 0.94)* | | 0.64 (0.41, 0.98)* |
| | Widowed | 0.41 (0.17, 1.01) | | 1.00 |
| Wealth index | Poorest | 1.00 | | 0.47 (0.19,1.15) |
| | Poorer | 0.89 (0.65, 1.23) | | 0.95 (0.69, 1.32) |
| | Middle | 0.78 (0.55,1.12) | | 0.88 (0.59, 1.32) |
| | Richer | 0.52 (0.36, 0.76)* | | 0.61 (0.39, 0.95)* |
| | Richest | 0.43 (0.29, 0.65)* | | 0.51 (0.31,0.82)* |
| Educational status | Uneducated | 1.00 | | 1.00 |
| | Primary | 0.95 (0.75, 1.19) | | 0.95 (0.77,1.26) |
| | Secondary and higher | 0.95 (0.70, 1.28) | | 0.88 (0.64,1.20) |
| Media exposure | No | 1.00 | | 1.00 |
| | Yes | 0.82 (0.66, 1.01) | | 0.85 (0.69,1.05) |
| Internet use | Never | 1.00 | | 1.00 |
| | In last 12 months | 0.62 (0.45, 0.86) | | 0.63 (0.45,0.87)* |
| | Before 12 months | 0.72 (0.37, 1.39) | | 0.75 (0.39, 1.46) |
| Health insurance | No | 1.00 | | 1.00 |
| | Yes | 0.68 (0.46, 1.01) | | 0.75 (0.50, 1.12) |
| **Community level factors** | | | | |
| Place of residence | Urban | | 1.00 | 1.00 |
| | Rural | | 0.99 (0.73, 1.35) | 0.99 (0.71, 1.36) |
| Region | Central Zone | | 1.00 | 1.00 |
| | Western Zone | | 1.09 (0.66, 1.80) | 1.08 (0.65, 1.79) |
| | Northern Zone | | 0.55 (0.44, 0.86) | 0.56 (0.35,0.89)* |
| | Southern highland Zone | | 0.45 (0.28, 0.71) | 0.42 (0.26, 0.68)* |
| | Southern Zone | | 1.17 (0.66, 2.09) | 1.05 (0.59, 1.91) |
| | Southwest highland zone | | 1.06 (0.66, 1.71) | 0.99 (0.59,1.91) |
| | Lake zone | | 0.77 (0.52, 1.15) | 0.74 (0.50, 1.12) |
| | Eastern Zone | | 1.16 (0.72, 1.87) | 1.10 (0.68, 1.79) |
| | Zanzibar | | 1.53 (0.33) | 1.47 (0.95, 2.28) |
| Distance to health facilities | Not big problem | | 1.00 | 1.00 |
| | Big problem | | 0.95 (0.76, 1.15) | 0.94 (0.77, 1.16) |
| Community illiteracy | Low illiteracy | | 1.00 | 1.00 |

(*Continued*)

**Table 3.** (Continued)

| | | Dropout from CoC AOR (95% CI) | | |
|---|---|---|---|---|
| **Variables** | **Categories** | **Model 1** | **Model 2** | **Model 3** |
| | | (Individual-level variables) | (Community-level variables) | (Both community and individual level variables) |
| | High illiteracy | | 0.91 (0.72, 1.17) | 0.89 (0.69, 1.15) |
| Community wealth index | Low poverty | | 1.00 | 1.00 |
| | High poverty | | 0.81 (0.60,1.09) | 0.99 (0.70, 1.39) |

*p-value > 0.05.

Women aged 15–19 exhibited increased odds of dropout from ANC compared to women in other age groups. Conversely, women in the 40–44 age group had lower odds of dropout from institutional delivery compared to their younger counterparts (15–19 years). This pattern is corroborated by studies in Ethiopia [29] and Mozambique [21]. This report could be associated with an increase in knowledge and decision-making abilities as women age [30]. Grand multiparous women showed an increased likelihood of dropout from ANC and institutional delivery, consistent with reports from other studies [18, 31]. This report may be linked to the preference of nulliparous women for giving birth in health facilities because of their heightened sensitivity to pregnancy-related complications [32].

Moreover, the likelihood of dropout from the maternal CoC was notably lower among married women and unmarried women exhibited increased odds of dropout from ANC compared to their married counterparts. This findings aligns with reports from studies conducted in Ethiopia [33] and Nepal [34], and may be attributed to lack of spousal support, as well as potential social isolation for unmarried pregnant women [35], factors that may contribute to a propensity to discontinue ANC visits and CoC.

Women with secondary and higher education demonstrated lower odds of dropout from institutional delivery compared to those with no education. This finding aligns with studies conducted elsewhere [18, 33, 36]. The educational attainment of women may enhance their decision-making abilities [37], potentially reducing the likelihood of dropout from institutional delivery. Furthermore, women with a higher wealth index demonstrated reduced odds of dropout from the CoC. This consistent trend is supported by studies conducted elsewhere [18, 33, 34]. The improved economic status of women might facilitate access to maternal CoC services in nearby facilities [37], potentially leading to a more seamless completion of the CoC.

In comparison to women with no internet exposure, those who used the internet in the past 12 months exhibited lower odds of dropout from maternal CoC. This aligns with a study in Sierra Leone [38]. The prevalence of internet usage, with its high interactivity and accessibility, particularly on sensitive subjects like reproductive health [39], is believed to play a pivotal role in assisting pregnant women in their decision-making during the early stages of pregnancy [40]. Similarly, women with health insurance showed reduced odds of dropout from the CoC, findings consistent with other study studies [41, 42]. This could be attributed to the role of health insurance in overcoming financial barriers that often impede access to maternal health service [43]. The odds of dropout from ANC and institutional delivery were more likely reported among women who had problems related to distance to health facility. This finding is supported by other studies [18, 44]. It is obvious that distance to health facility is considered as a barrier to utilize health services. Finally, geographic area emerged as a crucial factor, with the odds of dropout from CoC decreasing among women residing in certain geographic zones. This underscores the importance of geographic considerations in predicting the utilization of MNCH care services.

### Strengths and limitations of the study

This study delves into factors associated with dropout from maternal CoC, employing a multi-level binary logistic analysis to account for the hierarchical nature of the data. The use of nationally representative data strengthens the reliability and generalizability of the findings. However, relying on a secondary dataset prompted us to include institutional delivery as one component of the CoC rather than focusing specifically on skilled birth attendants, potentially overlooking distinctions between skilled birth attendants and institutional delivery.

### The implication of the study

This study may offer crucial insights for policymakers and stakeholders, providing valuable perspectives in order to navigate the landscape of maternal healthcare, fostering targeted and sustainable interventions, and driving sustainable improvements for the well-being of mothers and children in Tanzanian.

## Conclusion

The dropout rate from the maternal CoC in Tanzania was high. The findings contribute to our understanding of the complex dynamics surrounding maternal care continuity and underscore the need for targeted interventions, considering factors both at individual- and community-level.

## Supporting information

**S1 Table. Background characteristics of respondents and distribution of dropout from maternity CoC across independent variables among reproductive age women in Tanzania.** (DOCX)

**S2 Table. Random effect result and model comparison for the 3 maternal CoC.** (DOCX)

**S3 Table. Multilevel analysis of factors associated with dropout from the three maternal CoC among Tanzania reproductive age women.** (DOCX)

## Author Contributions

**Conceptualization:** Angwach Abrham Asnake, Amanuel Alemu Abajobir, Beminat Lemma Seifu, Yohannes Mekuria Negussie.

**Data curation:** Angwach Abrham Asnake, Amanuel Alemu Abajobir, Beminat Lemma Seifu, Molalgn Melese, Meklit Melaku Bezie, Yohannes Mekuria Negussie.

**Formal analysis:** Angwach Abrham Asnake.

**Methodology:** Angwach Abrham Asnake, Amanuel Alemu Abajobir, Beminat Lemma Seifu, Yordanos Sisay Asgedom, Molalgn Melese, Meklit Melaku Bezie, Yohannes Mekuria Negussie.

**Software:** Angwach Abrham Asnake.

**Validation:** Angwach Abrham Asnake, Yordanos Sisay Asgedom, Molalgn Melese, Meklit Melaku Bezie.

**Writing – original draft:** Angwach Abrham Asnake, Yohannes Mekuria Negussie.

**Writing – review & editing:** Angwach Abrham Asnake, Amanuel Alemu Abajobir, Beminat Lemma Seifu, Yordanos Sisay Asgedom.

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
