## [Decision Letter · Decision Letter 0]

19 Feb 2024

PONE-D-23-42201Multilevel analysis of dropout from maternal continuum of care and its associated factors in Tanzania: Evidence from 2022 Tanzania Demographic and Health Survey.PLOS ONE

Dear Dr. Asnake,

Thank you for submitting your manuscript to PLOS ONE. After careful consideration, we feel that it has merit but does not fully meet PLOS ONE’s publication criteria as it currently stands. Therefore, we invite you to submit a revised version of the manuscript that addresses the points raised during the review process.

We look forward to receiving your revised manuscript.

Kind regards,

Johanna Pruller, Ph.D.

Associate Editor

PLOS ONE

Journal Requirements:

**Additional Editor Comments:**

Please note that we have only been able to secure a single reviewer to assess your manuscript. We are issuing a decision on your manuscript at this point to prevent further delays in the evaluation of your manuscript. Please be aware that the editor who handles your revised manuscript might find it necessary to invite additional reviewers to assess this work once the revised manuscript is submitted. However, we will aim to proceed on the basis of this single review if possible. The reviewer has raised multiple minor concerns, in particular recommending to include more details in the introduction and methods section. Additionally, they are suggesting to improve the discussion, to reflect current literature findings. Could you please carefully revise the manuscript to address all comments raised?==============================

Reviewers' comments:

Reviewer's Responses to Questions

**Comments to the Author**

1. Is the manuscript technically sound, and do the data support the conclusions?

Reviewer #1: Yes

2. Has the statistical analysis been performed appropriately and rigorously? 

Reviewer #1: Yes

3. Have the authors made all data underlying the findings in their manuscript fully available?

Reviewer #1: Yes

4. Is the manuscript presented in an intelligible fashion and written in standard English?

Reviewer #1: Yes

5. Review Comments to the Author

Reviewer #1: 1. The introduction section should be better structured, and it should be explicitly stated that previous studies have shown that a very small proportion of women receive care throughout the CoC, and that many of them dropout in the CoC and therefore do not receive the completed care package, reducing the scope of health benefits that the CoC offers for the woman and her child.

2. The introduction should establish more clearly the concept of dropout of the CoC and the different moments in which it can be observed throughout the CoC. Likewise, in the available literature there is not a single definition for CoC dropout. On the contrary, there are restrictive definitions of the different types of dropouts for prenatal care, childbirth and postpartum. It is recommended to refer to it, its strengths, weaknesses, and the factors that have been associated with the results available in the evidence.

3. In the methodology section, it is suggested to include the way in which the independent variables were analysed at the individual and community levels. Likewise, for the latter, indicate whether they were also extracted with information from the same survey or from other sources of information.

4. In the results section, in Figure 1, the bars for the different moments of the CoC and for the general analysis do not add up to 100% except for the dropout for PNC. It is suggested to review and correct.

5. In the discussion section on lines 252-254 it is noted, “The overall rate of maternal CoC (83.86%) is lower than the rates reported in studies conducted in Ethiopia [19], Tanzania [18], Uganda [21], Gambia, and Ghana [24], and higher than the reports from a study conducted in Kenya [25]” However, in reality, according to Figure 1, this value It is suggested to review and correct. refers to the percentage of women who do not receive the full attention of the CoC continues.

6. It is suggested to add in the discussion section a reflection on how the findings of the present study contribute to the evidence that exists in the literature on the CoC linkages and what hypotheses the presented findings may arise in this sense for this country.

6. PLOS authors have the option to publish the peer review history of their article (what does this mean?). If published, this will include your full peer review and any attached files.

Reviewer #1: **Yes: **Ileana Beatriz Heredia Pi

---

## [Author Response · Author response to Decision Letter 0]

4 Mar 2024

Response to Reviewers 

Manuscript title: Multilevel analysis of dropout from maternal continuum of care and its associated factors in Tanzania: Evidence from 2022 Tanzania Demographic and Health Survey.

Manuscript ID: PONE-D-23-42201

Dear editor, 

PLOS ONE

We greatly value the valuable feedback provided by the reviewers. The constructive comments and suggestions from both the reviewers and editors offer significant opportunities to enhance the quality and clarity of the manuscript. We have carefully reviewed each comment and suggestion, and the subsequent pages include a comprehensive response addressing each point raised. Unmarked and marked-up copies of the revised manuscript are uploaded. Moreover, a response that responds to each point raised by the editor/reviewer(s) is highlighted below. 

Thank you for the opportunity to improve our manuscript based on your insights.

Sincerely,

Angwach Abrham on behalf of all authors

Point-by-point response for editor/reviewer comments 

Response to Reviewer-1

1: Introduction 

1. The introduction section should be better structured, and it should be explicitly stated that previous studies have shown that a very small proportion of women receive care throughout the CoC, and that many of them dropout in the CoC and therefore do not receive the completed care package, reducing the scope of health benefits that the CoC offers for the woman and her child.

Authors’ response: Thank you reviewer for your comment. We have duly addressed this (pp: 3-5). 

2. The introduction should establish more clearly the concept of dropout of the CoC and the different moments in which it can be observed throughout the CoC. Likewise, in the available literature there is not a single definition for CoC dropout. On the contrary, there are restrictive definitions of the different types of dropouts for prenatal care, childbirth and postpartum. It is recommended to refer to it, its strengths, weaknesses, and the factors that have been associated with the results available in the evidence.

Authors’ response: Thank you reviewer for your comment. We have duly addressed this (pp 4-5, line 77-86). 

2: Method 

3. In the methodology section, it is suggested to include the way in which the independent variables were analysed at the individual and community levels. Likewise, for the latter, indicate whether they were also extracted with information from the same survey or from other sources of information.

Authors’ response: Thank you reviewer for your comment. All variables included in this study were extracted from the same survey (2022 TDHS). All individual-level variables were taken from the survey except media exposure, which was generated by adding exposures to radio and/or television. Community-level variables (i.e. community illiteracy and community wealth index) were generated by aggregating individual-level variables into community-level variables. Other community-level variables (i.e. place of residence, geographic area, and distance to the health facility) were simply taken from the survey as it is. Overall, we have explicitly described these (please see revised manuscript; pp: 7-8, lines 139-147).

3: Results 

4. In the results section, in Figure 1, the bars for the different moments of the CoC and for the general analysis do not add up to 100% except for the dropout for PNC. It is suggested to review and correct.

Authors’ response: Thank you reviewer for your comment. We have corrected it in the revised Figure 1.

4: Discussion 

5. In the discussion section on lines 252-254 it is noted, “The overall rate of maternal CoC (83.86%) is lower than the rates reported in studies conducted in Ethiopia [19], Tanzania [18], Uganda [21], Gambia, and Ghana [24], and higher than the reports from a study conducted in Kenya [25]” However, in reality, according to Figure 1, this value It is suggested to review and correct. refers to the percentage of women who do not receive the full attention of the CoC continues.

Authors’ response: Thank you reviewer for your comment. We have revised and corrected it in the revised manuscript (pp 19, line 276).

6. It is suggested to add in the discussion section a reflection on how the findings of the present study contribute to the evidence that exists in the literature on the CoC linkages and what hypotheses the presented findings may arise in this sense for this country.

Authors’ response: Thank you reviewer for your suggestions. We have addressed it in the revised manuscript (pp: 19)

---

## [Decision Letter · Decision Letter 1]

10 Apr 2024

PONE-D-23-42201R1Multilevel analysis of dropout from maternal continuum of care and its associated factors: Evidence from 2022 Tanzania Demographic and Health Survey.PLOS ONE

Dear Dr. Asnake,

Thank you for submitting your manuscript to PLOS ONE. After careful consideration, we feel that it has merit but does not fully meet PLOS ONE’s publication criteria as it currently stands. Therefore, we invite you to submit a revised version of the manuscript that addresses the points raised during the review process.

Please submit your revised manuscript by May 25 2024 11:59PM. If you will need more time than this to complete your revisions, please reply to this message or contact the journal office at plosone@plos.org. Please include the following items when submitting your revised manuscript:A rebuttal letter that responds to each point raised by the academic editor and reviewer(s). You should upload this letter as a separate file labeled 'Response to Reviewers'.A marked-up copy of your manuscript that highlights changes made to the original version. You should upload this as a separate file labeled 'Revised Manuscript with Track Changes'.An unmarked version of your revised paper without tracked changes. You should upload this as a separate file labeled 'Manuscript'.If applicable, we recommend that you deposit your laboratory protocols in protocols.io to enhance the reproducibility of your results. Protocols.io assigns your protocol its own identifier (DOI) so that it can be cited independently in the future. For instructions, see: https://journals.plos.org/plosone/s/submission-guidelines#loc-laboratory-protocols. Additionally, PLOS ONE offers an option for publishing peer-reviewed Lab Protocol articles, which describe protocols hosted on protocols.io. Read more information on sharing protocols at https://plos.org/protocols?utm_medium=editorial-email&utm_source=authorletters&utm_campaign=protocols.

We look forward to receiving your revised manuscript.

Kind regards,

Birhan Tsegaw Taye

Academic Editor

PLOS ONE

Journal Requirements:

Additional Editor Comments:

1. Dear Authors, please revise your manuscript in detail and address each and every concern of the reviewers. Mainly, you are expected to clarify the methods used for this study, the description of the results section and the discussion section.

2. Please correct the country name 'Kenia' as Kenya, line 254 in the discussion section. 

3. I suggest that discussion section start with a paragraph highlighting the most relevant accomplishments of the research. This section can be improved as follows.

a) Main findings in the first paragraph

b) Comparison with existing literature

c) Strengths and Weaknesses of the study should be clearly highlighted

d) Conclusion

Reviewers' comments:

Reviewer's Responses to Questions

**Comments to the Author**

1. If the authors have adequately addressed your comments raised in a previous round of review and you feel that this manuscript is now acceptable for publication, you may indicate that here to bypass the “Comments to the Author” section, enter your conflict of interest statement in the “Confidential to Editor” section, and submit your "Accept" recommendation.

Reviewer #1: All comments have been addressed

Reviewer #2: All comments have been addressed

2. Is the manuscript technically sound, and do the data support the conclusions?

Reviewer #1: Yes

Reviewer #2: Yes

3. Has the statistical analysis been performed appropriately and rigorously? 

Reviewer #1: Yes

Reviewer #2: Yes

4. Have the authors made all the data underlying the findings in their manuscript fully available?

Reviewer #1: Yes

Reviewer #2: Yes

5. Is the manuscript presented in an intelligible fashion and written in standard English?

Reviewer #1: Yes

Reviewer #2: Yes

6. Review Comments to the Author

Reviewer #1: (No Response)

Reviewer #2: You did a splendid job and I only have some comments and questions, probably:

Result

Check the paragraph starting @ line 209,

its not clear to start with and the numbers indicating dropouts of ANC are incongruent. 27.13% vs (82.62%).

Analysis of dropout to ANC, institutional delivery and PNC

If you have to analyze each components of the CoC, then what is the value of merging them to one variable (CoC)? And the same applies to your discussion, you need to have a good justification to have a question of continuum of care dropout and discuss about a single area of care dropout - ANC or other components.

Discussion

It is recommended to start a discussion by stating your purpose of the study and then revealing the major findings.

3rd paragraph (line 287-93)

Your justifications for older mothers nullified your justification for grand multiparity all in a single paragraph. It needs revision, I think.

4th paragraph (line 294)

Odds of dropouts of CoC are lower among married women and unmarried women have higher odds of dropout for ANC - your results stated and I am not clear how these statements goes different. They look the same but you stated otherwise.

7. PLOS authors have the option to publish the peer review history of their article (what does this mean?). If published, this will include your full peer review and any attached files.

Reviewer #1: **Yes: **Ileana Beatriz Heredia Pi

Reviewer #2: No

---

## [Author Response · Author response to Decision Letter 1]

12 Apr 2024

Response to Reviewers 

Manuscript title: Multilevel analysis of dropout from maternal continuum of care and its associated factors in Tanzania: Evidence from 2022 Tanzania Demographic and Health Survey.

Manuscript ID: PONE-D-23-42201R1

Dear editor, 

PLOS ONE

We greatly value the valuable feedback provided by the editor and reviewers. These constructive comments and suggestions offer significant opportunities to enhance the quality and clarity of the manuscript. We have carefully reviewed each comment and suggestion, and indicated point-by-point comprehensive responses addressing each point raised on subsequent pages. Moreover, unmarked and marked-up copies of the revised manuscript are uploaded. 

Thank you for the opportunity to improve our manuscript based on your insights.

Sincerely,

Angwach Abrham on behalf of all authors

Point-by-point response for editor/reviewer comments 

Response to Editor-

1. Dear Authors, please revise your manuscript in detail and address each and every concern of the reviewers. Mainly, you are expected to clarify the methods used for this study, the description of the results section and the discussion section.

Authors’ response: Thank you editor for your suggestions. We have carefully revised these in respective sections.

2. Please correct the country name 'Kenia' as Kenya, line 254 in the discussion section. 

Authors’ response: Thank you for your comment. This was a typo; we have corrected it. 

3. I suggest that discussion section start with a paragraph highlighting the most relevant accomplishments of the research. This section can be improved as follows.

a) Main findings in the first paragraph

b) Comparison with existing literature

c) Strengths and Weaknesses of the study should be clearly highlighted

d) Conclusion

Authors’ response: Thank you very much for this suggestion. We have addressed it in the revised manuscript (Discussion section). 

 Response to Reviewer-

1: Results

1. Check the paragraph starting @ line 209,

its not clear to start with and the numbers indicating dropouts of ANC are incongruent. 27.13% vs (82.62%).

Authors’ response: Thank you reviewer for your careful consideration. This was an editorial issue; we have rephrased it accordingly. 

2. Analysis of dropout to ANC, institutional delivery and PNC

If you have to analyze each components of the CoC, then what is the value of merging them to one variable (CoC)? And the same applies to your discussion, you need to have a good justification to have a question of continuum of care dropout and discuss about a single area of care dropout - ANC or other components.

Authors’ response: Thank you reviewer for your comments. Incorporation of all these 3 components in the formulation of the CoC variable ensures that essential preventive care services are comprehensively administered during these critical points of contact (Introduction: lines 98-104; Methods: lines 125-127; 135-136; 142-144). Moreover, COC outlines important connections in time (when the care is given) and place (where the care is given). To progress toward higher coverage of crucial and evidence-based interventions, women stay healthy, and newborns have the best start in life when they receive high-quality care during pregnancy, delivery, and the early postnatal period. Analyzing each component of the CoC separately is important to identify which component of CoC had high dropout and to minimize the health cost by intervening on the most common dropout. So, we supported the overall CoC findings with each component of the continuum. 

2: Discussion

1. It is recommended to start a discussion by stating your purpose of the study and then revealing the major findings.

Authors’ response: Thank you for your comments. We have addressed this in the revised manuscript (lines 261-266)

2. Your justifications for older mothers nullified your justification for grand multiparity all in a single paragraph. It needs revision, I think.

Authors’ response: Thank you so much for your significant insight. We have addressed it in the revised manuscript.

3. 4th paragraph (line 294)

Odds of dropouts of CoC are lower among married women and unmarried women have higher odds of dropout for ANC - your results stated and I am not clear how these statements goes different. They look the same but you stated otherwise.

Authors’ response: Thank you very much for your comments and suggestions. We have rephrased it.

---

## [Editor Report · Decision Letter 2]

17 Apr 2024

Multilevel analysis of dropout from maternal continuum of care and its associated factors: Evidence from 2022 Tanzania Demographic and Health Survey.

PONE-D-23-42201R2

Dear Dr. Asnake,

We’re pleased to inform you that your manuscript has been judged scientifically suitable for publication and will be formally accepted for publication once it meets all outstanding technical requirements.

Within one week, you’ll receive an email detailing the required amendments. When these have been addressed, you’ll receive a formal acceptance letter, and your manuscript will be scheduled for publication.

An invoice will be generated when your article is formally accepted. Please note that if your institution has a publishing partnership with PLOS and your article meets the relevant criteria, all or part of your publication costs will be covered. Please make sure your user information is up-to-date by logging into Editorial Manager at Editorial Manager® and clicking the ‘Update My Information' link at the top of the page. If you have any questions relating to publication charges, please contact our author billing department directly at authorbilling@plos.org.

If your institution or institutions have a press office, please notify them about your upcoming paper to help maximize its impact. If they’ll be preparing press materials, please inform our press team as soon as possible—no later than 48 hours after receiving the formal acceptance. Your manuscript will remain under a strict press embargo until 2 pm Eastern Time on the date of publication. For more information, please contact onepress@plos.org.

Kind regards,

Birhan Tsegaw Taye

Academic Editor

PLOS ONE